# Lipids in Atherosclerosis: Pathophysiology and the Role of Calculated Lipid Indices in Assessing Cardiovascular Risk in Patients with Hyperlipidemia

**DOI:** 10.3390/ijms24010075

**Published:** 2022-12-21

**Authors:** Melania Gaggini, Francesca Gorini, Cristina Vassalle

**Affiliations:** 1Institute of Clinical Physiology, National Research Council, Via Moruzzi 1, 56124 Pisa, Italy; 2Fondazione CNR—Regione Toscana G Monasterio, Via Moruzzi 1, 56124 Pisa, Italy

**Keywords:** atherosclerosis, lipids, non-invasive lipid indices, non-HDL, CRI-I, CRI-II, TRL-C, AIP

## Abstract

The role of lipids is essential in any phase of the atherosclerotic process, which is considered a chronic lipid-related and inflammatory condition. The traditional lipid profile (including the evaluation of total cholesterol, triglycerides, high-density lipoprotein, and low-density lipoprotein) is a well-established tool to assess the risk of atherosclerosis and as such has been widely used as a pillar of cardiovascular disease prevention and as a target of pharmacological treatments in clinical practice over the last decades. However, other non-traditional lipids have emerged as possible alternative predictors of cardiometabolic risk in addition to traditional single or panel lipids, as they better reflect the overall interaction between lipid/lipoprotein fractions. Therefore, this review deals with the lipid involvement characterizing the pathophysiology of atherosclerosis, discussing some recently proposed non-traditional lipid indices and, in the light of available knowledge, their actual potential as new additive tools to better stratify cardiovascular risk in patients with hyperlipidemia as well as possible therapeutic targets in the clinical practice.

## 1. Introduction

Coronary artery disease (CAD), the main clinical manifestation of atherosclerosis, still represents the main cause of mortality and morbidity in both sexes all over the world [1]. Nonetheless, in the course of atherosclerosis, not only does ischemic heart disease develop but also cerebrovascular disease and peripheral arterial disease [2]. Moreover, it is worth noting that endovascular procedures play a very important role in the treatment of atherosclerotic diseases, but the process of restenosis limits their effectiveness and contributes to the need for re-intervention [3,4].

Long considered a degenerative disease mainly determined by a passive accumulation of lipids, atherosclerosis has been subsequently demonstrated as an inflammatory disease characterized by lipid accumulation, chronic low-grade inflammation, and endothelial dysfunction and involving oxidative modified lipoprotein infiltration, immune cell activation, and extracellular matrix changes, with evidence of lipids as key players and/or regulators of these events [5,6,7,8]. In particular, the traditional lipid profile (total cholesterol—TC, triglycerides—TG, high-density lipoprotein cholesterol—HDL, and low-density lipoprotein cholesterol—LDL) has always been considered an essential tool for the assessment of cardiovascular disease (CVD) prevention and treatment in clinical practice. However, other non-traditional lipids and indices have been proposed, some of them showing an even greater predictive role for CVD and ischemic stroke than traditional single lipid parameters [9,10].

Surely, the role of these non-traditional lipid tools in relation to atherosclerosis and risk assessment still needs to be further evaluated, in particular, to assess their routine applicability in clinical practice and optimal cut-offs and identify patients that would benefit most from their use. This review briefly describes the lipid involvement in the cellular and molecular mechanisms characterizing the pathophysiology of all atherosclerosis phases. Furthermore, the purpose of the present review is to discuss some recently proposed non-traditional lipid indices simply calculated using the traditional lipid profile; in the light of available knowledge, the state of the art before their introduction into clinical practice for the routine management and prevention of atherosclerosis; and what aspects still need to be further improved in this field.

## 2. Key Role of Lipids in Atherosclerosis

The two main processes involved in the pathogenesis of atherosclerosis include cholesterol deposition and chronic inflammation [11]. In particular, according to the lipid theory of atherosclerosis, lipid peroxidation and the oxidation of LDL trigger initiation and further progression of atherosclerosis [11,12]. The main transporter of cholesterol to target cells is LDL, a heterogeneous class of lipoprotein particles consisting of a hydrophobic core containing TG and cholesterol esters in a hydrophilic surface membrane of phospholipids, free cholesterol, and apolipoproteins (principally ApoB-100), the latter mediating the binding of LDL particles to specific cell-surface receptors [13,14]. Modified LDL appear to be a major causative agent in the atherosclerotic process by stimulating endothelial cells (EC) to produce inflammatory markers with consequent cytotoxic effects; inhibiting nitric oxide (NO)-induced vasodilatation; and promoting the recruitment of monocytes to the vessels, macrophage progression to foam cells, and the migration and proliferation of vascular smooth muscle cells (VSMC) [15,16,17] (Table 1). Although oxidized low-density lipoproteins (ox-LDL) have long been considered the only type of modified LDL crucial for atherogenesis, at least three modified LDL forms (i.e., small dense, electronegative, and desialylated LDL), have been detected in the bloodstreams of atherosclerosis patients [18,19]. These molecules act as factors able to stimulate LDL aggregation, LDL association with the extracellular matrix components, and the formation of LDL-containing immune complexes, and all of them are susceptible to oxidation by resident vascular cells [13,20,21,22]. In particular, the small dense subfraction is characterized by an enhanced atherogenicity due to its increased susceptibility to modifications and its high binding affinity to the proteoglycans contained in the intima layer of the arterial wall, and desialylated LDL exhibit enhanced uptake and a low degradation rate once internalized, while electronegative LDL show a high propensity for self-association [19,23].

In the initiation of the atherosclerotic process, modified lipoproteins accumulated in the intima activate the endothelium [20]. Furthermore, the reduced expression of endothelial NO synthase and superoxide dismutase, which are responsible for maintaining an effective barrier and reducing oxidative stress, respectively, affects endothelial barrier integrity and determines the retention of atherogenic LDL. Transcriptional activation of nuclear factor kappa B promotes the production of cytokines (e.g., tumor necrosis factor-alpha—TNF-α; interleukin–IL-1, IL-4, IL-6; interferon gamma—IFN-γ), which, in turn, induces the expression of monocytes and leukocyte adhesion molecules (such as vascular cell adhesion molecule-1, intercellular adhesion molecule-1, and E-selectin on the endothelial surface) and leads to the migration of monocytes and lymphocytes into the inner arterial wall [11,24,25]. Monocytes differentiate into macrophages [25] that internalize excess lipids derived from modified LDL, resulting in the intracellular accumulation of cholesterol esters, generation of foam cells (the hallmark of atherosclerotic lesion), and the aggregation of foam cells to form fatty streaks, the earliest grossly recognizable atherosclerotic lesions [11,22] (Table 1).

Although macrophages represent the major infiltrating cells, adaptive immunity, which is modulated by T and B cells, and some effector cells such as mast cells and eosinophils play a central role in the advancement and expansion of atherosclerosis through the secretion of cytokines (IL-6, IFN-γ) and high-affinity IgA, IgE, and IgG antibodies [26]. Monocytes are also capable of differentiating into dendritic cells, a type of leukocyte that contain an elevated content of LDL-receptor 1 (LOX-1) and significantly contribute to ox-LDL uptake [27,28]. VSMC are also implicated in the development of the atherosclerotic plaque and in the transition from a fatty streak to a fibrous fatty lesion [22,29]. After their migration from the medial to the intima vascular layers, they proliferate in response to platelet-derived growth factor and basic fibroblast growth factor secreted by EC and macrophages, respectively, and produce extracellular matrix molecules such as collagen and elastin, which form the atherosclerotic plaque cap (Table 1) [22,29,30]. The ox-LDL–LOX-1 interaction, in addition to supporting the migration and proliferation of VSMC, may also promote their apoptosis and the release of matrix-degrading enzymes (i.e., metalloproteinases—MMPs 1 and 9) [22,27]. VSMC migration leads to the generation of the atheromatous fibrous caps that enclose a lipid-rich necrotic core, and their thickness, cellularity, matrix composition, and collagen content determine the characteristics and vulnerability of the atherosclerotic plaque (Table 1) [31]. Instead, calcification may occur in advanced plaque progression lesions, more frequently found in elderly subjects, where microcalcifications characterize a phase of the unstable plaque, while strong, dense calcification generally reflects a more stable plaque (Table 1) [31,32]. During the progression of plaque development, macrophages and T lymphocytes produce proteolytic enzymes, which may induce cup rupture, a coagulation process, and blood clot and lead to clinical events (Table 1) [33,34].

As endothelial dysfunction takes part as a critical step in atherosclerosis onset and development, it is worth mentioning that flow-mediated dilation is the most important method for endothelial dysfunction assessment in the literature as well as in clinical practice, although not without limitations (e.g., poor standardization and requirement of well-trained, experienced operators, aspects which limit reproducibility) [35,36].

Among biomarkers, in view of the pivotal role of lipids in the pathogenesis and progression of the atheromatous plaque, the traditional lipid panel (including TC, TG, HDL, and LDL) has long been identified as useful for assessing the risk factor of atherosclerosis and has been widely used as a pillar for cardiovascular (CV) disease prevention and treatment in clinical practice over the last decades [8,37,38,39]. However, other non-traditional lipids have emerged as possible alternative predictors of cardiometabolic risk in addition to traditional single or panel lipids, better reflecting the overall interaction between lipid/lipoprotein fractions [40,41,42].

## 3. Oxidized LDL in Assessing CV Risk and Non-Invasive Lipid Indices

Atherogenic dyslipidemia is characterized by high serum TG, low concentrations of HDL, and high concentrations of small dense LDL. In particular, the relationship between higher serum levels of LDL and an increased risk of CV disease is the basis for the management of dyslipidemia in the current guidelines, which recommended LDL as the primary target of CV risk prevention [43]. However, although CV risk is substantially reduced in patients by the LDL-lowering effects of statins, these individuals still have a residual CV risk, as some subjects who reach their LDL target may be subject to the adverse effects of other biomarkers of the lipoprotein-lipid profile [44]. Accordingly, the reduction of CV risk could be enhanced by combination therapy that aims to decrease elevated LDL and other lipids (for now, mainly TG).

For this reason, other non-traditional lipid indices have been proposed and evaluated and could be used as possible additive or alternative predictors of cardiometabolic risk in addition to the traditional single or panel of traditional lipids in the routine management of dyslipidemia and for the prevention of atherosclerosis and its complications in clinical practice (in order to further stratify patients and better identify those at high risk). Among these biomarkers, one interesting parameter because of its key role in atherosclerosis onset and development is ox-LDL. The generation of these atherogenic particles derived through lipoprotein oxidation under oxidative stress is one of the earliest events in the pathogenesis of atherosclerosis. Ox-LDL promote a number of adverse events that mediate endothelium damage and mark the course of the atherosclerotic plaque up to its rupture and acute clinical events, including the following: (1) adherence and accumulation of monocytes and adhesion and aggregation of platelets; (2) release of chemokines and growth factors; (3) oxidative stress and reactive oxygen species production; (4) impairment of NO release and function; (5) endoplasmic reticulum stress; (6) apoptosis; and (7) thrombus rupture [45,46,47].

In addition to the large amount of basic research evidencing the key role of ox-LDL in the pathogenesis of atherosclerosis, a number of studies and a relatively recent meta-analysis (including three nested case-control studies, one case-cohort study, five hospital-based cohort studies, and three community-based cohort studies) indicated that increased levels of ox-LDL are associated with CV events in studies, while another meta-analysis documented the beneficial effects of statin on circulating ox-LDL levels [48,49,50]. Ox-LDL was found to have higher discriminatory power to predict cardiometabolic risk than LDL [51,52,53].

The evaluation of antibodies to ox-LDL may also represent an additive tool to improve CV risk stratification and the reclassification of patients within risk categories, and a consistent amount of data on the clinical correlates of anti-ox-LDL antibodies is available [54].

Nonetheless, as ox-LDL is a mixture of heterogeneously modified particles, several ELISA assays are available, the differential use of which may introduce high levels of heterogeneity between the obtained results, making the comparison of data difficult. Moreover, the oxidation of LDL can be affected by a number of different factors (e.g., obesity, triglycerides, systemic inflammation, LDL particle size, diet, exercise, smoking habit, and pharmacological therapy), which could be considered in the future prior to the introduction of ox-LDL in routine patient assessments [50]. On the other hand, performing any adjunctive test requires qualified staff and additive time and costs.

Interestingly, nitrated lipoproteins, produced by the nitration of the tyrosyl residues of apolipoproteins by myeloperoxidase, are emerging as a potential lipid biomarker associated with cardiovascular risk, with more available data on nitrated HDL than on nitrated LDL particles at present [55]. In this context, the nitration of HDL particles appears related to a decrease in the activity of caspase-3, paraoxonase-1, and cholesterol transport via ABCA1, which decrease the cardiovascular benefits characterizing the HDL particles [55].

Moreover, some non-traditional lipid indices, i.e., biomarkers easily calculated using the traditional lipid panel at no cost (practically always available as a part of the patient screening panel) such as non-HDL cholesterol (non-HDL), Castelli risk index I (CRI-I), Castelli risk index II (CRI-II), triglyceride-rich lipoprotein cholesterol (TRL-C), and atherogenic index of plasma (AIP), have been proposed and discussed beyond the new alternative lipid parameters as potential additive tools in assessing CV risk in patients with hyperlipidemia.

### 3.1. Non-HDL Cholesterol

Non-HDL is a biomarker reflecting the amount of cholesterol content within atherogenic lipoproteins, which have a critical role in atherosclerosis risk through multiple mechanisms. In particular, these effects may be mediated by the level of ApoB-containing particles, and/or the vascular deposition of cholesterol ester-enriched smaller TRL eliciting pro-inflammatory/thrombotic events, and/or their association with dysfunctional proatherogenic HDL particles [56,57,58,59]. This lipid parameter may be easily calculated by subtracting the HDL value from total plasma cholesterol and thus can be obtained from a standard lipid panel without requiring additional time or testing costs (Table 2) [60].

Epidemiological studies have documented the relationship between higher non-HDL and increased ischemic risk [61,62]. Accordingly, very recent data have suggested that non-HDL may serve as a reliable predictor of recurrent ischemic stroke and all-cause death at one-year follow-up in patients with acute ischemic stroke [63]. Previous results showed that non-HDL concentration is significantly associated with an increased risk of cardiac death in diabetic patients in the Third National Health and Nutrition Examination Survey (n = 1,122,299 deaths; median follow-up 12.4 years) [64]. In particular, a meta-analysis including seven studies (448,732 subjects belonging to the general population) evidenced an increased CAD risk (more pronounced in men than in women) for individuals in the highest non-HDL category [65]. Moreover, the Framingham Offspring Study indicated that elevated levels of non-HDL are related to a higher risk of mortality [66]. In the Framingham Offspring Cohort, subjects who had a low 10-year CV risk and exhibited non-HDL ≥160 mg/dL over an 11- to 20-year period had a stronger association with CV events (CAD, myocardial infarction—MI, angina, death) compared with non–HDL below that threshold or with a shorter time period of exposure to non-HDL levels above 160 mg/dL [67]. These results were confirmed by another meta-analysis evaluating 11,057 subjects with CAD (six trials, follow-up range from 18 to 148 months). The authors reported that increased levels of non-HDL are significantly associated with an increased risk of mortality, indicating that this biomarker may represent an easy and reliable tool for assessing adverse outcomes in the general population as well as in patients with CV disease [68]. Furthermore, subjects with high non-HDL levels had a higher CV risk independently of their LDL levels, and non-HDL was proven superior in its predictive value of major CV events when compared to LDL [53,65,69]. Moreover, recent results reported that non-HDL and TC-to-HDL ratio are significantly correlated to arterial stiffness levels, even when levels of LDL are low (<70 mg/dL), suggesting that these biomarkers are related to residual CV risk [70]. Consistently, non-HDL and TC/HDL (CRI-I) have also shown predictive value for the residual risk of acute MI/sudden death in men with LDL levels <120 mg/dL [71]. Interestingly, the EPIC (European Prospective Investigation Into Cancer and Nutrition) study reported that even in the group of subjects with LDL <100 mg/dL, participants with high non–HDL levels, high TG levels, or a high TC-to-HDL ratio had an increased risk for CV disease [40]. Moreover, in the Framingham Study cohort, non-HDL was a stronger predictor of CV risk than LDL, even when the analysis was repeated considering TG levels [72]. In fact, non-HDL, along with ApoB (a component of several types of lipoproteins, such as the blood transporters of cholesterol very low density lipoproteins—VLDL, intermediate-density lipoprotein—IDL, and LDL), are indicated as more reliable biomarkers than LDL in hypertriglyceridemic individuals and in subjects with a very low LDL concentration [73].

As increasing results have suggested that non-HDL (and ApoB) are better than LDL at predicting CV risk, these biomarkers have been proposed as secondary targets in some treatment guidelines [74]. At present, some pharmacological agents seem effectively reliable for lowering the circulating levels of these biomarkers, although their capacity to reduce residual CV risk is still under evaluation in several ongoing clinical trials [75,76].

LDL, non-HDL, and ApoB appear generally highly correlated, therefore, they provide similar information on CV risk [77]. Nonetheless, in certain groups of subjects (e.g., patients with elevated TG levels, type 2 diabetes—T2D, obesity, or very low LDL levels), the value of LDL (obtained through calculation or directly measured) may underestimate the total concentration of cholesterol carried by LDL and the total concentration of ApoB-containing lipoproteins, thereby underestimating the risk of CV disease, whereas the measurement of non-HDL (or ApoB) is advised [77]. On the other hand, non-HDL and ApoB, although highly correlated, do not provide the same information, as non-HDL does not give any indication of the number of particles, which is instead reflected by ApoB because one ApoB molecule is carried on the surface of each atherogenic lipid particle [78].

Findings from intervention trials have revealed that any strategy that lowers the non-HDL concentration in parallel also reduces the incidence of ischemic heart disease [61,62,79]. In particular, a meta-analysis focusing on the relationship between non-HDL reduction and CAD risk showed that most lipid-modifying drugs used as monotherapy (e.g., statins, resins, fibrates, or niacin) retain a 1:1 relationship, which means that for each 1% lowering of non-HDL, CAD risk decreases by 1% [80]. In this context, there are interesting data regarding rosuvastatin at moderate- and high-intensity doses and simvastatin and atorvastatin at high-intensity doses (for the prevention of CAD in people with T2D), recently reported as effective in lowering non-HDL [81].

The European ESC/EAS 2019 guidelines for the management of dyslipidemia advocate non-HDL evaluation for risk assessment, particularly in people with high TG levels, T2D, obesity, or very low LDL levels, with a Class I recommendation (recommended or indicated), Level C (consensus of opinion of the experts and/or small studies, retrospective studies, registries), and set non-HDL target of <2.2 mmol/L (<85 mg/dL), <2.6 mmol/L (<100 mg/dL), and <3.4 mmol/L (<130 mg/dL) in people with very high, high, and moderate CV risk, respectively (Table 2) [77]). Moreover, non-HDL is endorsed by the updated guidelines of the European Society of Cardiology on CV disease prevention in clinical practice (with representatives of the European Society of Cardiology and 12 medical societies and the special contribution of the European Association of Preventive Cardiology) [37]. Although LDL remains the primary treatment target of dyslipidemia, these clinical guidelines also focus on modifications in TRL, such as non-HDL and ApoB, whose evaluation is highly recommended [37]. Specifically, non-HDL is included as a factor for the Systemic Coronary Risk Estimation 2 (SCORE2, the algorithm used to estimate the 10-year risk of death from CVD) and SCORE2-Older Persons (SCORE2-OP, which estimates an individual’s 10-year risk of fatal and non-fatal CV events—MI, stroke—in apparently healthy subjects between 40–69 years with untreated or stable risk factors for several years) risk algorithms.

Data from 2516 participants of the Framingham Offspring Study (25–40 years of age, free of CV disease and T2D, mean follow-up 32.6 years) showed that the trend of lipid concentration remained generally stable over the 30-year life course, whereas the mean non-HDL evaluated in young adults was highly predictive of the values observed later in life [82]. Consequently, each subject could be classified as high or low non-HDL on the basis of two measurements collected between 25 and 40 years of age. Interestingly, 80% of those with elevated non-HDL (≥160 mg/dL) at the first two samplings remained in the high group over time (25 years), while 88% of those with low non-HDL (<130 mg/dL) remained in the low group (non-HDL below 160 mg/dL) [82]. The risk of CV disease in individuals with high non-HDL in young adulthood was 22.6% vs. a risk of 6.4% in subjects in the low non-HDL group [82]. Hence, lipid monitoring before 40 years of age could help to identify the individuals most exposed to elevated lipid levels and with a long-term high CV risk, opening an opportunity for potential more aggressive and targeted preventive lipid-lowering strategies during early midlife.

These findings were confirmed by other research in which non-HDL levels were measured at three life stages, adolescence (2–18 years), young adulthood (21–30 years), and mid-adulthood (33–45 years), indicating that all life stages are associated with coronary atherosclerosis in mid-adulthood (with non-HDL in adolescents having the highest association with the presence of CAD in mid-adulthood) with important implications for CV prevention [83].

### 3.2. Castelli Index I and II

CRI-I, obtained by calculating the TC-to-HDL ratio, has a close relationship with CV risk in both sexes [84]. Optimal values of CRI-I are defined as <5 and <4.5 for men and women, respectively (Table 2). In particular, subjects with TC/HDL greater than or equal to 4.5 had a higher CV risk in the Framingham Study [65]. The Castelli risk index II (CRI-II), demonstrated as a good indicator of CV risk, is calculated as the LDL-to-HDL ratio, with ideal values of <3.5 and 3 in men and women, respectively (Table 2) [84,85]. Therefore, CV risk is higher as the ratios increase, reflecting the imbalance between the cholesterol carried by atherogenic (numerator) and protective (denominator) lipoproteins for an increase in the proatherogenic part expressed in the numerator, a decrease in the anti-atherosclerotic component reported in the denominator, or both trends. The high correlation between the two Castelli indices is probably because most of the cholesterol is contained in LDL, thus TC and LDL are closely related. Likewise, changes in both indices are shown to be better predictors for CV risk reduction when compared to changes in individual lipids or lipoproteins [85].

As suggested in a population-based prospective study from Norway comprising 33,859 individuals, with 2746 individuals diagnosed with MI during follow-up, CRI-I is a strong independent predictor of MI in men [86]. Previously, in the older Rancho Bernardo cohort study (1386 women and 1094 men), only CRI-I was able to predict coronary heart disease and CV disease deaths in women independent of estrogen use and other risk factors [87]. The ATTICA study, conducted during 2001–2012 on 1514 men and 1528 women, revealed that CRI-I has a high prognostic significance even in the general population, although this association is stronger in women, thus representing a simple tool for assessing individual long-term CV risk and, therefore, timely interventions [88]. Recently, we compared indices of cardiometabolic risk in overweight/obese workers by sex, observing that CRI-I and CRI-II were significantly higher in males than in females. Additionally, both biomarkers correlated with waist circumference, body mass index (BMI), blood pressure, and fasting glucose, although more frequently in females, while among males only homocysteine was correlated with CRI-I. These sex-related differences in the association between CV risk factors and insulin resistance (IR)/cardiometabolic indices could be a potential pathophysiological determinant of the observed epidemiological sex-related differences in cardiometabolic diseases and deserve further study in the future [89].

Notably, in subjects who present triglyceridemia over 300 mg/dL (3.36 mmol/L), the calculation of LDL is not reliable, and the use of TC/HDL is more adequate. As an example, in the Quebec Cardiovascular Study, CRI-I and CRI-II were calculated to understand which index best predicts the risk of ischemic heart disease in overweight hyperinsulinemic patients with high-TG/low-HDL dyslipidemia. The study focused on the serum components of the two indices because there is more cholesterol in the VLDL fraction in individuals with higher TG concentrations, and CRI-II may underestimate the magnitude of the dyslipidemic state in these patients [90]. Therefore, although both LDL/HDL and TC/HDL significantly correlate with the characteristics of the atherogenic metabolic status related to the IR syndrome (elevated levels of insulin, ApoB, and small dense LDL particles), TC/HDL appears to work better in these patients.

CRI-I and CRI-II were also evaluated in 120 Indian subjects divided into two groups: the case group included 60 angiographically confirmed patients with CAD, and the control group consisted of 60 age- and sex-matched healthy volunteers. Serum TC and LDL did not show any significant difference between the two groups; in particular, serum TC was around 200 mg/dL in 74% of patients, and LDL was below 130 mg/dL in 86% of patients with CAD. Although the serum values of the individual components of the indices were not different in the two groups, CRI-I and CRI-II significantly differed between the patients with CAD and the healthy subjects, helping to stratify patients and identify the subjects with dyslipidemia who are at CV risk [91].

In a previous study that explored the relationship between an elevated CRI-I value (≥4) and proximal CAD in a group of individuals undergoing multislice computed tomography, subjects with CRI-I ≥ 4 compared to those with a ratio <4 had a higher prevalence of proximal plaque (62% vs. 48%) and significant CAD (19% vs. 9%). In a multivariate logistic regression analysis, only age, sex, and CRI-I ≥ 4 were associated with significant CAD and proximal plaque, and patients with CRI-I ≥ 4 had a higher frequency of obstructive CAD and atherosclerotic plaque, despite statin use [92]. Therefore, in view of the lack of costs and wide availability for adult subjects, the authors proposed the measurement of the TC/HDL ratio as a screening biomarker to identify individuals who might benefit from more aggressive preventive therapies (e.g., lifestyle changes and/or lipid-lowering therapy) when the index is elevated.

### 3.3. Triglyceride-Rich Lipoprotein Cholesterol

TRL-C is obtained by subtracting HDL and LDL from TC, an alternative and simple way to calculate the so-called remnant cholesterol (remnant-C) indicating the cholesterol component of TG-rich lipoproteins (Table 2). This biomarker has been associated with cardiometabolic risk [93]. Specifically, circulating levels of TG and cholesterol transported in TRL can predict CV events, with TRL-C ≥ 30 mg/dL identifying subjects at a higher CV risk [94]. Accordingly, the association of TG and remnant cholesterol with major adverse cardiac events—MACEs (MI, stroke, or CV death)—was evaluated in a cohort of older individuals at high CV risk in the PREDIMED study, a randomized controlled trial that examined the effects of the Mediterranean diet compared with a low-fat diet for the primary prevention of CV disease in high-risk subjects. TRL-C was distributed in the same manner between intervention groups (Mediterranean diet enriched with extra-virgin olive oil and Mediterranean diet enriched with nuts) and between sexes and was increased with increasing BMI in subjects with diabetes compared with those without diabetes. Furthermore, regardless of LDL concentration, TRL-C ≥ 30 mg/dL was able to differentiate subjects at a higher risk of MACEs compared with TRL-C at a lower concentration, indicating TRL-C as a useful biomarker for assessing residual lipid risk for MACEs in subjects at high CV risk but who had no previous CV disease [94]. Previously, within two diverse, prospective, longitudinal observational US cohorts including nearly 5000 participants overall, TRL-C was shown to be associated with a 23% increased risk of incident CAD for the combined population during an 8-year follow-up, after adjustment for CV risk factors. Nonetheless, this association was attenuated after adjustment for HDL and LDL in the model [95]. TRL-C was also evaluated in patients with CAD and LDL <100 mg/dL following lipid-lowering therapy. When a stepwise multivariate Cox proportional hazard analysis was conducted, TRL-C was a significant predictor of CV events after adjustment for TG, non-HDL, and total ApoB, resulting in a superior performance to non-HDL in predicting CV events in CAD patients with LDL levels <100 mg/dL who are undergoing lipid-lowering treatment. Thus, this biomarker might serve as a key target to reduce residual risk after LDL goals are reached through lipid-lowering therapy [96].

By examining 97,962 participants from the Copenhagen City Heart Study and the Copenhagen General Population Study combined, which were followed prospectively for up to 22 years, higher levels of TRL-C and LDL resulted in equal associations with increased risk of CAD and MI; however, only TRL-C concentration was associated with increased all-cause mortality risk, with hazard ratios ranging from 1.0 (for 19.3–38.2 mg/dL TRL-C) to 1.6 (≥58 mg/dL TRL-C) [97]. Furthermore, increased concentrations of TRL-C but not higher concentrations of LDL were associated with increased all-cause mortality in 5414 Danish patients diagnosed with CAD [98]. Very recently, Liu et al. (2021) assessed the baseline high-sensitivity C-reactive protein (hsCRP) in 6723 CAD patients with recurrent CV events and TG <2.3 mmol/L to explore the association between TRL-C and recurrent cardiovascular events (RCVEs) and whether this relationship may depend on systemic inflammation in statin-treated patients with CAD and with normal TG levels [99]. The risk of RCVEs was investigated across quartiles of baseline TRL-C and stratified according to the median of hsCRP. The authors reported that the highest quartile of TRL-C was significantly associated with a risk of RCVEs even after stratification of hsCRP, whereas adjusting for age, sex, and LDL levels did not affect this association, evidencing that this biomarker could be a reliable marker of risk stratification and a treatment target in this patient population [99].

Overall, the growing body of observational evidence suggests that an increased concentration of remnant cholesterol may cause atherosclerosis, similar to increased LDL, because of its capability to penetrate the atrial wall and to be retained within the arterial intima, ultimately leading to the accumulation of cholesterol and development of atherosclerotic plaque [100]. In addition, TG-rich lipoproteins may both upregulate proinflammatory cytokines, adhesion molecules, and prothrombotic factors in ECs and enhance the recruitment and activation of monocytes, promoting the development of foam cells [93,101,102].

### 3.4. Atherogenic Index of Plasma

The index AIP, calculated as the logarithmically transformed ratio of molar concentrations of TG to HDL, not only represents a reliable predictor of cardiometabolic risk, reflecting the relationship between protective and atherogenic lipoproteins, but also represents a strong predictor of atherosclerosis and CAD [103,104]. In particular, AIP values < 0.11 are associated with a low CV risk (Table 2), and recent studies have indicated that AIP is progressively emerging as a powerful and reliable predictor of cardiometabolic risk in the general population, e.g., increased risk for developing CVD in postmenopausal women; independent risk factor for CAD in subjects undergoing coronary angiography; and significantly associated with MetS, hypertension and T2D, and adverse prognosis in patients with acute MI [103,104,105,106]. Indeed, a cross-sectional study performed on a total number of 194 women in the age group from 30 to 60 years showed that AIP is positively and significantly correlated with age, BMI, systolic blood pressure, and diastolic blood pressure [104]. In a hospital-based case-control study conducted in China on 2936 CAD patients and 2451 controls, AIP was positively and significantly correlated with TC, TG, LDL, non-HDL, TC/HDL, and LDL/HDL, and the most strongly lipid parameter associated with CAD in the univariate analysis and an independent determinant for CAD in the multivariate analysis [105]. Shin et al. (2022) showed that among 1292 adult men who participated in the Korea National Health and Nutrition Examination Survey, AIP increased with the increase in obesity, blood glucose, and blood lipid profile, suggesting that it could predict cardiometabolic risk [107]. Moreover, Li et al. (2018) reported that in 2523 patients with T2D without lipid-lowering treatments, AIP was independently related to waist circumference, HOMA-IR, fasting plasma glucose, systolic blood pressure, and uric acid [108]. These findings were confirmed in a 9-year longitudinal study conducted with Taiwanese citizens that showed that AIP was able to identify the high-risk subjects of both genders, especially in the middle-aged group in subjects having MetS, hypertension, and T2D [102]. On the other hand, we recently reported that in overweight/obese middle-aged subjects, AIP was significantly higher in males than in females. In addition, AIP was correlated with waist circumference, BMI, blood pressure, fasting glucose, CRP, and fibrinogen in females but only with homocysteine in males, suggesting the possibility of a different relationship with cardiometabolic factors among the two sexes [89].

The prognosis of T2D individuals who underwent percutaneous coronary intervention was significantly worse in the presence of high levels of AIP compared to that of the low-AIP group, and the effect of this index on prognosis was not affected by LDL [109]. According to Zhou and co-authors (2021), AIP is not only the lipid parameter that most strongly correlates with incident CAD in patients with T2D, but it is also best suited for the T2D population of all the many nonconventional indices, thus providing a useful tool for the effective prevention of CV complications in these patients [110].

Very recent studies have suggested that AIP and TC/HDL-C may represent reliable markers of residual risk in patients with CAD or suspected CAD, even for subjects who did not present any traditional CV risk factor [111,112].

AIP was also significantly higher in Chinese patients with acute coronary syndrome (N = 376) compared to the control group [113]. In contrast, the incidence of all-cause mortality in hospitalized MI patients was significantly higher in the low-AIP group (<0.24) than in the high-AIP group (>0.24), with low AIP predicting all-cause mortality independently with a significant risk ratio of 3.71 in the multivariate analysis [106]. Thus, further insights are needed to clarify the precise relationship between this biomarker and outcomes in this patient population.

## 4. Conclusions

It is universally accepted that the pathogenesis of atherosclerosis is closely related to lipids, from LDL uptake by monocytes and macrophages and accumulation in the arterial intima to their involvement in inflammation. Lipid-lowering therapies (e.g., statins) are critical to prevent and manage CV disease, and the evaluation of the traditional lipid profile (TC, TG, HDL, and LDL) remains the cornerstone diagnostic test for dyslipidemia and its management and for CV risk assessment [77]. However, as dyslipidemia is a dynamic field of research, increased knowledge in the lipid world is further required to develop the best in-clinic practices. Therefore, in addition to the conventional lipid panel, other non-traditional lipid-related biomarkers have been studied and have emerged as being significantly correlated with cardiometabolic risk and disease, among which biomarkers non-HDL is the most investigated and the only calculated index recognized as an effective biomarker and is, as such, reported in different international society guidelines (together with ApoB). For example, the 2019 ESC/EAS Guidelines for the management of dyslipidemia recommend a non-HDL evaluation of CV risk estimation in subjects with high TG levels, T2D, obesity, or very low LDL levels (Class I recommended or is indicated; level of evidence C/consensus of opinion of the experts and/or small studies, retrospective studies, registries) [77]. In particular, the residual risk among statin-treated subjects may be reduced by the evaluation of non-HDL (or ApoB but, in this case, adjunctive costs may be considered). Thus, the available guidelines advise the use of non-HDL-C (or ApoB) as a secondary treatment target in high-risk or very high risk patients with mild-to-moderate hypertriglyceridemia [77]. Interestingly, a meta-analysis (including 12 independent reports for a total of 233,455 subjects and 22,950 events) estimated that a non-HDL-C strategy would prevent 300,000 more events than an LDL-C strategy in the US adult cohort over a 10-year statin treatment period [114]. Thus, a reasonable approach could be the performance of the traditional lipid profile together with the evaluation of non-HDL, which can be calculated at no additional cost. The main strengths and specific shortcomings of the use of non-HDL for the evaluation of CV risk estimation in dyslipidemia are summarized in Table 3. Notably, the evaluation of remnant-C may be more useful in some cases than non-HDL, as non-HDL is not able to differentiate between LDL and remnant-C. Specifically, some subjects may present a high remnant-C despite a low LDL, and thus a relatively low non-HDL, so the interpretation of non-HDL without considering remnant-C may be misleading [115]. Nonetheless, it should be taken into account that if LDL is calculated using the Friedewald equation, remnant-C (corresponding to TG/2.2 in mmol/L or TG/5 in mg/dL) does not provide any clinical information in addition to TG; different if directly measured LDL is used in the formula.

Interestingly, increasing evidence suggests that calculated non-traditional serum lipid parameters (i.e., TC/HDL beyond non-HDL) may better predict CV risk when compared to LDL, and they could serve as auxiliary tools to assess residual risk in cardiometabolic patients [9,40,44,116,117].

In any case, the role of these non-traditional calculated lipid tools in relation to atherosclerosis and risk assessment still needs to be further evaluated, in particular, to assess their routine applicability in clinical practice and optimal cut-offs and identify the patients that would benefit the most from their use.

Notably, these indices, however, may not work for all cases. One example can be represented by carriers of Apolipoprotein A-I (Milano) or (Paris) (ApoA-I[Milano] and ApoA-I[Paris]), rare cysteine ApoA-I variants that lead to HDL deficiency despite reduced CV risk, a status that makes these indices inapplicable to these patients [118,119]. Additionally, TRL-C is easily obtainable from available laboratory parameters, except for severe hypertriglyceridemia forms where the estimation of LDL using the Friedewald formula is inapplicable.

Therefore, in general, for non-traditional calculated lipids, the main advantages include the fact that their values can be easily derived from a few lipid biomarkers routinely tested in the overall patient evaluation and, as such, are universally available without any supplemental costs for sanitary systems (Figure 1). However, it can also be observed that reference intervals, decision values, and cut-offs for non-traditional lipid indices are not univocally defined. As age and sex can represent important confounding factors, the possibility to determine the reliability of specific cut-offs according to these variables should be tested [89,120]. Moreover, we recently evidenced a poor concordance between some of these lipid indices when used to identify subjects at higher cardiometabolic risk [89]. In fact, these indices have different pathophysiological bases and/or different correlations with CV risk factors [89]. Consequently, as each lipid parameter may measure different physiopathological aspects of dysglycemia and cardiometabolic risk and give different information, they should not be used interchangeably in daily clinical practice without caution (Figure 1). On the other hand, it is this characteristic of incomplete overlap that also may represent an advantage, as different lipid indices may provide incremental information that, when combined, could lead to a more accurate prediction of risk and outcome (Figure 1).

These calculated parameters, although promising for CV risk assessment and retaining unquestionable advantages (including availability, ease of calculation, and lack of additive costs), still need wider use to better compare and interpret results and further verification in different patient populations to precisely establish their significance in different clinical settings before these indices can be spread and incorporated into routine prevention and treatment decisions. Further validation also remains partially mandatory for non-HDL, the best choice among them [121].

In conclusion, it is important to remember that CVD is a very complex condition. Thus, a better understanding of the molecular characteristics of these indices based on the lipid components used for their calculation, which have different peculiarities, may help to further stratify specific groups of patients using a point of view of “stratified medicine” in order to refine and reduce the residual risk, connecting bench to bedside and vice versa (Figure 2).

## Figures and Tables

**Figure 1 ijms-24-00075-f001:**
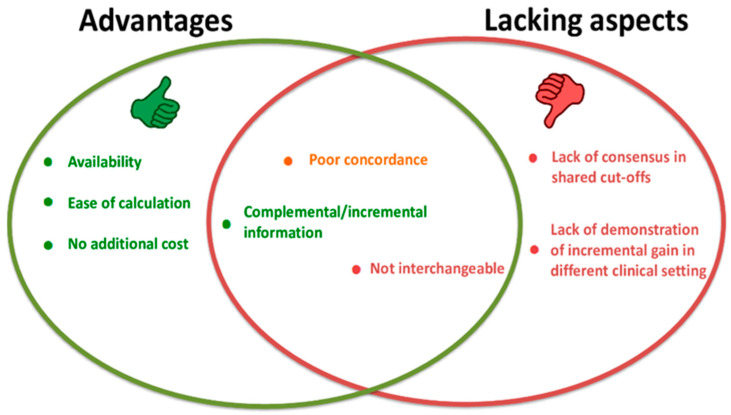
Main advantages/disadvantages of the new proposed non-invasive lipid indices.

**Figure 2 ijms-24-00075-f002:**
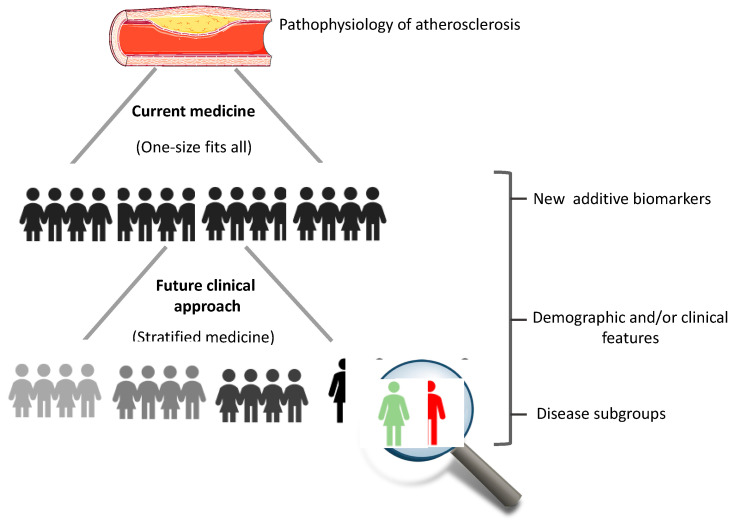
Evolution from current to future medicine. Stratification and revised classification towards a more personalized approach to clinical practice should occur through a growing understanding of molecular and biological mechanisms.

**Table 1 ijms-24-00075-t001:** Summary of the critical molecules and events characterizing the main phases in the onset and development of the atherosclerotic plaques.

Endothelial Dysfunction	Fatty Streak	Fibrous Plaque	Vulnerable Plaque	Plaque Erosion, Rupture, Thrombosis
- Endothelial activation - Upregulation of adhesion molecules - Increased vascular permeability - Monocyte and other cell recruitment and infiltration - Impaired vasodilation (reduced NO) - ROS and inflammatory mediators - LDL intake/accumulation in the arterial intima	- VSMC proliferation - Lipid accumulation - Foam cell development - T lymphocyte infiltration - Platelet aggregation - Macrophage activation - Inflammation	- Deposition of extracellular connective tissue matrix - Fibrous cap formation - Lipid-laden foam cell core containing lipid necrotic debris and calcium - Vascular remodeling - Luminal narrowing - Flow abnormalities - Reduced oxygen supply inflammatory mediated	- Proteolytic enzymes (MMPs) - Intraplaque hemorrhage - Thinning of the fibrous cap - Microcalcification - Necrotic-rich lipid core (apoptosis and necrosis) - Inflammatory mediators	- Platelet aggregation - Fibrin polymerization - Inflammatory coagulation and proteolysis mediators - Thrombosis - Acute coronary syndrome
			**Calcific plaque** Dense calcification deposition	

Abbreviations: LDL—low-density lipoprotein; MMPs—metalloproteinases; NO—nitric oxide; ROS—reactive oxygen species; VSMC—vascular smooth muscle cells.

**Table 2 ijms-24-00075-t002:** Calculated lipid indices, their calculation, and suggested cut-offs.

Index (Acronym)	Formula	Threshold Values (mg/dL)
Non-HDL Cholesterol (Non-HDL-C)	Total cholesterol (mg/dL)–High-density lipoproteins (mg/dL)	<130
Castelli risk index 1 (CRI-I)	Total cholesterol (mg/dL)/High-density lipoproteins (mg/dL)	Males: <5; females <4.5
Castelli risk index 2 (CRI-II)	Low-density lipoproteins (mg/dL)/High-density lipoproteins (mg/dL)	Males: <3.5; females <3
Triglyceride-Rich Lipoprotein Cholesterol (TRL-C)	Non-HDL–Low-density lipoproteins (mg/dL)	<30
Atherogenic index of plasma (AIP)	Log (Triglycerides (mg/dL)/High-density lipoproteins (mg/dL)	≤0.11

**Table 3 ijms-24-00075-t003:** Specific strengths and aspects to improve in the use of non-HDL for evaluation of CV risk estimation in dyslipidemia.

Strengths	Shortcomings
Independence from TG levels	Lack of distinction between remnant-C and LDL
Easily available, high throughput, and fast turnaround time	Arbitrary risk cut-offs
Calculation in the nonfasting state	Dependency from HDL measurement errors in hypertriglyceridemia, which may influence the calculation of non-HDL
Inclusion of remnant-C	Better identification of confounding factors, interferences
No additional cost above conventional lipid testing in terms of time and assay	Lack of familiarity for most practitioners
Additive utility beyond existing markers (residual risk)	
Advised when LDL is low or TG are increased	
Associations with CV risk and treatment target (risk reduction proportional to the degree of non-HDL lowering)	

Abbreviations: HDL—high-density lipoproteins; LDL—low-density lipoproteins; remnant-C—remnant cholesterol; non-HDL—non-HDL cholesterol; TG—triglycerides.

## Data Availability

Not applicable.

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
