# Peer review of "Lipids in Atherosclerosis: Pathophysiology and the Role of Calculated Lipid Indices in Assessing Cardiovascular Risk in Patients with Hyperlipidemia"

_ijms, 2022, doi:10.3390/ijms24010075_

Round 1

Reviewer 1 Report (Previous Reviewer 1)

The authors improved the manuscript considerably.

Author Response

Dear Editor,

Please find enclosed the revised manuscript “Lipids in atherosclerosis: pathophysiology and the role of calculated lipid indices in assessing cardiovascular risk in patients with hyperlipidemia” by Gaggini M et al. for publication in IJMS.

In the new version, we addressed all reviewers' comments and modified the text accordingly. For details, please refer to the point-by-point responses to reviewers attached hereby. Please, also consider the great effort for the revision requiring extensive reorganization and re-writing of the manuscript, with the addition of two new tables and more than 20 references.

We wish to thank the Editor and Reviewers for their constructive appreciation and comments, and we hope that the manuscript in the current form is suitable for publication.

Thanking you for your kind attention and consideration, we look forward to hearing from you.

Yours sincerely,

Dr. Cristina Vassalle

Reviewer 1

The authors described the non-traditional lipid indices as possible diagnostic tools for coronary artery disease in the clinical practice.

The topic is interesting and it is within the scope of the journal and there are some novelties. However, there are important remarks that should be addressed.

Major remarks

1.- The introduction is redundant with the section “Lipids in atherosclerosis”. Authors should improve the introduction section to better present the structure of the main text and avoid redundant information.

The introduction section was rewritten and shortened to facilitate reading and main text presentation and avoid redundant parts

2.- If the aim of the manuscript is to promote the use of the non-traditional lipid indices as additional diagnostic tools and as possible therapeutic targets in the clinical practice, Authors should include a table summarizing the meaning of each index and patients to whom each index is useful according to their particular characteristics.

These issues were better discussed in the text in general and a table reporting specific strengths and flaws in the use of non-HDL (the most suitable and shared biomarker for the evaluation of CV risk estimation in dyslipidemia) added as Table 3.

3.- How would these indices be interpreted in carriers of apo AI Milano or apo AI Paris?

This relationship was discussed as follow in the text:

“Notably, these indices, however, may not work for all cases. One example can be represented by carriers of Apolipoprotein A-I (Milano) or (Paris) (ApoA-I(Milano) and ApoA-I(Paris)), rare cysteine ApoA-I variants that lead to HDL deficiency despite reduced CV risk, status that makes these indices inapplicable in these patients (112, 113). Additionally, TRL-C is easily obtainable by deriving from available laboratory parameters, except for severe hypertriglyceridemia forms where the estimation of LDL using the Frieldwald formula is inapplicable.”

4.- Is there some specific relationship between cardiovascular residual risk and the non-traditional lipid indices?

More specifical information on the relationship between indices and CV residual risk was added and discussed in the specific index sections

5.- Check the accuracy of the following sentences:

  1. a) line 174 “including TRLs in VLDL and their remnants”; VLDL are TRLs (triglycerides rich lipoproteins).

Checked and modified

  1. b) line 328, LDLs are not lipoproteins rich in triglycerides.

Checked and modified

Reviewer 2 Report (New Reviewer)

I have received to review the review article entitled “Lipids in atherosclerosis: pathophysiology and the role of calculated lipid indices in assessing cardiovascular risk in patients with hyperlipidemia”, prepared by Melania Gaggini et al. submitted to the International Journal of Molecular Sciences (IF=6.208). Cardiovascular diseases (CVDs) in the course of atherosclerosis are the leading cause of morbidity and mortality worldwide. Research related to increasing the knowledge about pathogenesis of CVDs are therefore of crucial importance. Lipid disorders belong to the most important risk factors for atherosclerosis development.

In my opinion, the manuscript represents high quality and it should be considered for publication in the future but some improvements are necessary which I listed below.

1)     The introduction is far too short and laconic. It should be noted that in the course of atherosclerosis, not only ischemic heart disease develops, but also cerebrovascular disease and peripheral arterial disease. It is worth noting that endovascular procedures play a very important role in the treatment of atherosclerotic diseases, but the process of restenosis limits their effectiveness and contributes to the need for re-intervention. In the introduction it has been mentioned that endothelial dysfunction takes a part in the atherosclerosis development. In my opinion, it should be worth mentioning that flow-mediated dilation is the most important method for endothelial dysfunction assessment in the research as well as in the clinical practice. Nitrated lipoproteins also should be mentioned as considered and discussed to be a potential lipid biomarker associated with cardiovascular risk. (10.3390/ijerph182211970; 10.3390/ijerph191811242)

2)     The purpose of the study should be precisely described at the end of the introduction.

3)     In my opinion it should be not “interleukin IL-1,4,6”, but “selected interleukins (IL-1, IL-4, and IL-6). (line 71)

4)     The text of the paper should be revised in terms of possible minor stylistic and editorial errors.

Round 2

Reviewer 2 Report (New Reviewer)

The paper has been significantly improved. I have no further comments. The paper is of high value and it has been prepared carefully.

This manuscript is a resubmission of an earlier submission. The following is a list of the peer review reports and author responses from that submission.

Round 1

Reviewer 1 Report

The authors described the non-traditional lipid indices as possible diagnostic tools for coronary artery disease in the clinical practice.

The topic is interesting and it is within the scope of the journal and there are some novelties. However, there are important remarks that should be addressed.  

Major remarks

1.- The introduction is redundant with the section “Lipids in atherosclerosis”. Authors should improve the introduction section to better present the structure of the main text and avoid redundant information.

2.- If the aim of the manuscript is to promote the use of the non-traditional lipid indices as additional diagnostic tools and as possible therapeutic targets in the clinical practice, Authors should include a table summarizing the meaning of each index and patients to whom each index is useful according to their particular characteristics.

3.- How would these indices be interpreted in carriers of apo AI Milano or apo AI Paris?

4.- Is there some specific relationship between cardiovascular residual risk and the non-traditional lipid indices?

5.- Check the accuracy of the following sentences:

a) line 174 “including TRLs in VLDL and their remnants”; VLDL are TRLs (triglycerides rich lipoproteins).

b) line 328, LDLs are not lipoproteins rich in triglycerides.

Author Response

Pisa, September 19, 2022

Dear Editor,

Please find enclosed the revised manuscript “Lipids in atherosclerosis: pathophysiology and the role of calculated lipid indices in assessing cardiovascular risk in patients with hyperlipidemia” by Gaggini M et al. for publication in IJMS.

In the new version, we addressed all reviewers' comments and modified the text accordingly. For details, please refer to the point-by-point responses to reviewers attached hereby. Please, also consider the great effort for the revision requiring extensive reorganization and re-writing of the manuscript, with the addition of two new tables and more than 20 references.

We wish to thank the Editor and Reviewers for their constructive appreciation and comments, and we hope that the manuscript in the current form is suitable for publication.

Thanking you for your kind attention and consideration, we look forward to hearing from you.

Yours sincerely,

Dr. Cristina Vassalle

Reviewer 1

The authors described the non-traditional lipid indices as possible diagnostic tools for coronary artery disease in the clinical practice.

The topic is interesting and it is within the scope of the journal and there are some novelties. However, there are important remarks that should be addressed.

Major remarks

1.- The introduction is redundant with the section “Lipids in atherosclerosis”. Authors should improve the introduction section to better present the structure of the main text and avoid redundant information.

The introduction section was rewritten and shortened to facilitate reading and main text presentation and avoid redundant parts

2.- If the aim of the manuscript is to promote the use of the non-traditional lipid indices as additional diagnostic tools and as possible therapeutic targets in the clinical practice, Authors should include a table summarizing the meaning of each index and patients to whom each index is useful according to their particular characteristics.

These issues were better discussed in the text in general and a table reporting specific strengths and flaws in the use of non-HDL (the most suitable and shared biomarker for the evaluation of CV risk estimation in dyslipidemia) added as Table 3.

3.- How would these indices be interpreted in carriers of apo AI Milano or apo AI Paris?

This relationship was discussed as follow in the text:

“Notably, these indices, however, may not work for all cases. One example can be represented by carriers of Apolipoprotein A-I (Milano) or (Paris) (ApoA-I(Milano) and ApoA-I(Paris)), rare cysteine ApoA-I variants that lead to HDL deficiency despite reduced CV risk, status that makes these indices inapplicable in these patients (112, 113). Additionally, TRL-C is easily obtainable by deriving from available laboratory parameters, except for severe hypertriglyceridemia forms where the estimation of LDL using the Frieldwald formula is inapplicable.”

4.- Is there some specific relationship between cardiovascular residual risk and the non-traditional lipid indices?

More specifical information on the relationship between indices and CV residual risk was added and discussed in the specific index sections

5.- Check the accuracy of the following sentences:

  1. a) line 174 “including TRLs in VLDL and their remnants”; VLDL are TRLs (triglycerides rich lipoproteins).

Checked and modified

  1. b) line 328, LDLs are not lipoproteins rich in triglycerides.

Checked and modified

Reviewer 2 Report

Dear authors

the work is only based on the description of alternative non 

Invasive lipid indices that are not new biomarkers deriving from the use of 

of alterative molecules, they are just new calculations starting from values already used

for the same application.

The work is too poor To be consider as a review

Author Response

Pisa, September 19, 2022

Dear Editor,

Please find enclosed the revised manuscript “Lipids in atherosclerosis: pathophysiology and the role of calculated lipid indices in assessing cardiovascular risk in patients with hyperlipidemia” by Gaggini M et al. for publication in IJMS.

In the new version, we addressed all reviewers' comments and modified the text accordingly. For details, please refer to the point-by-point responses to reviewers attached hereby. Please, also consider the great effort for the revision requiring extensive reorganization and re-writing of the manuscript, with the addition of two new tables and more than 20 references.

We wish to thank the Editor and Reviewers for their constructive appreciation and comments, and we hope that the manuscript in the current form is suitable for publication.

Thanking you for your kind attention and consideration, we look forward to hearing from you.

Yours sincerely,

Dr. Cristina Vassalle

Dear authors

the work is only based on the description of alternative non Invasive lipid indices that are not new biomarkers deriving from the use of alterative molecules, they are just new calculations starting from values already used for the same application.

The value of these indices precisely lies in their capacity to be derived by simple calculation from biomarkers already used for the same scope, but with increased capacity of patient risk stratification and correlation with residual risk, combining availability with added utility. As required by the reviewer 3, a part related to ox-LDL and their evaluation was added in Section 3.

The work is too poor To be consider as a review.

Please, consider the great effort for the revision requiring extensive reorganization and re-writing of the manuscript in the new version, with the addition of two new tables and more than 20 references.

Reviewer 3 Report

The review describes the pathogenesis of atherosclerotic plaque, the role of lipids in this process, and the importance of lipid indices in assessing the risk of coronary artery disease.

The authors presented an interesting work, but there are some comments on it:

1. Sections 2 and 3 are not very related to each other, since most of the atherosclerosis biomarkers listed in Table 1 are not included in the lipid indices from Section 3. It may be worth shortening Section 2 or removing it completely.

2. The authors should describe in Section 3 the role of oxidized LDL in assessing CHD risk.

3. The authors should insert a table containing all indexes described in the article and their threshold values.

4. The authors should slightly add to the title of the review (for example): «Lipid in atherosclerosis: pathophysiology and the role of non-invasive lipid indices in assessing the risk of coronary heart disease».

Author Response

Pisa, September 19, 2022

Dear Editor,

Please find enclosed the revised manuscript “Lipids in atherosclerosis: pathophysiology and the role of calculated lipid indices in assessing cardiovascular risk in patients with hyperlipidemia” by Gaggini M et al. for publication in IJMS.

In the new version, we addressed all reviewers' comments and modified the text accordingly. For details, please refer to the point-by-point responses to reviewers attached hereby. Please, also consider the great effort for the revision requiring extensive reorganization and re-writing of the manuscript, with the addition of two new tables and more than 20 references.

We wish to thank the Editor and Reviewers for their constructive appreciation and comments, and we hope that the manuscript in the current form is suitable for publication.

Thanking you for your kind attention and consideration, we look forward to hearing from you.

Yours sincerely,

Dr. Cristina Vassalle

Reviewer 3

The review describes the pathogenesis of atherosclerotic plaque, the role of lipids in this process, and the importance of lipid indices in assessing the risk of coronary artery disease.

The authors presented an interesting work, but there are some comments on it:

  1. Sections 2 and 3 are not very related to each other, since most of the atherosclerosis biomarkers listed in Table 1 are not included in the lipid indices from Section 3. It may be worth shortening Section 2 or removing it completely.

Part 2 and 3 were rewritten and reorganized to facilitate reading.

  1. The authors should describe in Section 3 the role of oxidized LDL in assessing CHD risk.

According to the reviewer’s suggestions, the section on oxLDL and their role in CHD risk was added in Section 3.

  1. The authors should insert a table containing all indexes described in the article and their threshold values.

Added as Table 2, a Table 3 was also added which reports strengths and aspects to improve in the use of non-HDL, the best choice index, for the evaluation of CV risk estimation in dyslipidemia.

  1. The authors should slightly add to the title of the review (for example): «Lipid in atherosclerosis: pathophysiology and the role of non-invasive lipid indices in assessing the risk of coronary heart disease».

The title was changed according to the reviewer’s suggestion as follows:

“Lipids in atherosclerosis: pathophysiology and the role of calculated lipid indices in assessing cardiovascular risk in patients with hyperlipidemia”

Round 2

Reviewer 3 Report

My comments have been taken into account. I don't have any new comments.